# Modifications of Thermal-Induced Northern Pike (*Esox lucius*) Liver Ferritin on Structural and Self-Assembly Properties

**DOI:** 10.3390/foods11192987

**Published:** 2022-09-25

**Authors:** Siying Zhang, Xin Guo, Xiaorong Deng, Yunfeng Zhao, Xinrong Zhu, Jian Zhang

**Affiliations:** School of Food Science and Technology, Shihezi University, Shihezi 832003, China

**Keywords:** northern pike, liver ferritin, thermal treatment, structure, self-assembly

## Abstract

Ferritin, as an iron storage protein, regulates iron metabolism and delivers bioactive substances. It has been regarded as a safe, new type of natural iron supplement, with high bioavailability. In this paper, we extracted and purified ferritin from northern pike liver (NPLF). The aggregation stabilities, assemble properties, and structural changes in NPLF were investigated using electrophoresis, dynamic light scattering (DLS), circular dichroism (CD), UV–Visible absorption spectroscopy, fluorescence spectroscopy, and transmission electron microscopy (TEM) under various thermal treatments. The solubility, iron concentration, and monodispersity of NPLF all decreased as the temperature increased, and macromolecular aggregates developed. At 60 °C and 70 °C, the α-helix content of ferritin was greater. The content of α-helix were reduced to 8.10% and 1.90% at 90 °C and 100 °C, respectively, indicating the protein structure became loose and lost its self-assembly ability. Furthermore, when treated below 80 °C, NPLF maintained a complete cage-like shape, according to the microstructure. Partially unfolded structures reassembled into tiny aggregates at 80 °C. These findings suggest that mild thermal treatment (80 °C) might inhibit ferritin aggregation while leaving its self-assembly capacity unaffected. Thus, this study provides a theoretical basis for the processing and use of NPLF.

## 1. Introduction

Iron deficiency is the most common nutritional deficiency and global public nutritional disorder [1]. The poor bioavailability of iron or insufficient supply of absorbable iron may cause iron deficiency anemia (IDA). At present, ferrous sulfate, sodium iron ethylenediamine tetraacetate (NaFeEDTA), and ferrous gluconate are used in oral iron preparations to mitigate IDA [2,3]. However, these supplements have adverse side effects, including unfavorable sensory experiences such as an unpleasant smell; gastrointestinal problems, such as constipation, diarrhea, nausea, and vomiting; and growth retardation of patients [4]. In addition, these supplements must be taken for a long time. To compensate for the aforementioned shortcomings, ferritin has been proposed as a new iron supplement with high iron absorption and low side effects [5]. 

Ferritin is a kind of iron storage protein that consists of a spherical protein shell and an iron core. It exists extensively in animals, plants, and bacteria. Ferritins in animals are mainly found in metabolically active tissues, such as the liver, spleen, and heart. In plants, ferritins are mainly found in the non-green plastids such as plastid, amyloid, and the top of the root [6]. The structures of ferritins are conservative, regardless of bacterial, plant ferritin, or animal origin. This special protein consists of 24 subunits. The outer diameter of the protein shell is 12.5 nm, and the diameter of the internal cavity is 8 nm. Thousands of Fe (III) exist in the inner cavity of ferritin as ferric iron hydroxide phosphate [5]. When iron concentrations in the environment are low, ferritin reduces Fe (III) to Fe (II) with the help of reductants (NADH/FMN/FADH2) and releases Fe (II) from inside the ferritin for use by the body [7,8,9,10]. In addition, plant ferritin can withstand higher temperatures and extreme pH conditions [5]. Previous studies have shown that ferritin has low sensitivity to various iron chelators in the diet. Moreover, ferritin has a specific receptor (AP-2) in small intestinal epithelial cells, which can be directly absorbed in a complete structure and transported to the cell interior [11]. Beard et al. administered iron in equal amounts as ferrous sulfate, soybean ferritin, and horse spleen ferritin to anemic rats [12]. After 21 d of treatment, the hemoglobin and hematocrit of anemic rats reached control values, indicating good bioavailability of ferritin. Iron absorption has been assessed from purified soybean ferritin in a human study [13]. Nonanemic women with low iron status were fed a standardized meal (bagel, cream cheese, and apple juice) containing 1 μCi Fe/meal as soybean ferritin or as ferrous sulfate. After 21 d, the absorption of iron was not significantly different (means of 30% and 34%, respectively) between soybean ferritin and ferrous sulfate groups. Thus, iron can be absorbed effectively from ferritin, and ferritin should be explored as a safe and effective functional iron supplement. It has been reported that protein structures are sensitive to environmental conditions, especially temperature. Therefore, the denaturation of protein at excessive temperature will limit its application in the food industry. Currently, the effect of thermal treatment on protein structural and physicochemical properties has attracted much attention. The research showed that the secondary and tertiary structure of protein can be changed during heat treatment, which can lead to the formation of protein aggregates [14]. Chang et al. reported that the sulfhydryl content, α-helix content, and the solubility of myofibrillar protein from mirror carp decreased with increasing temperature, which indicated that temperature can induce protein aggregation [15]. The monomer conformation of β-lactoglobulin was rearranged at 135–140 °C, which led to the formation of aggregates and changes in the structure of protein [16,17]. Tang et al. showed that thermal treatment (60–80 °C) prolonged the storage time of pea seed ferritin, inhibited the protein aggregation, and promoted its monodispersity [18]. However, treatment at higher temperatures (90 °C or higher) led to a change in ferritin structure. Li et al. described an increase in the particle size distribution and changes to the secondary and tertiary structures changed in recombinant oyster ferritin with increasing temperatures [19]. 

Recent studies showed that animal ferritin did not degrade during storage, due to the lack of extension peptide (EP), compared with plant ferritin. Thus, animal ferritin might be a better iron supplement than plant ferritin [20]. Currently, there are many studies on the thermal stability of plant ferritin and mammal ferritin, but few studies have reported on ferritin in aquatic products, especially fish liver. Northern pike (*Esox lucius*) liver (NPLF) is usually discarded as processing waste, but the liver contains ferritins. Therefore, the livers can still be utilized. The study of NPLF is helpful to reduce waste and improve the comprehensive utilization of northern pike liver. 

In this paper, we purified ferritin from the liver of northern pike, and we investigated the effects of thermal treatment on the structural and self-assembly properties of ferritin in order to optimize the thermal treatment conditions of ferritin for applications in food systems.

## 2. Materials and Methods

### 2.1. Materials and Chemicals

*Esox lucius* (weight 900 ± 50 g, length 50 ± 5 cm, *n* = 15) samples were purchased from a local market of aquatic products (Shihezi, China). The fish were sent to the laboratory using plastic bags containing suitable oxygen within 30 min and then stunned instantly through a blow to the head with a wooden stick. The livers were collected after euthanization. Then the livers were homogenized with a crusher (Scientz Co., Ningbo, China), freeze-dried, and stored at –80 °C until extraction. Acrylamide, 2-hydroxy-1-ethanethiol, sodium lauryl sulfate (SDS), tetramethyl ethylenediamine (TEMED), ammonium persulfate (APS), ethylenediamine tetra acetic acid (EDTA), and tris (hydroxymethyl) methyl aminomethane were purchased from Beijing Biotopped, Ltd. (Beijing, China). Ammonium thioglycolate was purchased from Shanghai Macklin, Ltd. (Shanghai, China).

### 2.2. Purification of Ferritin

The extraction of ferritin was performed using the method reported by Li et al. [21], with some modifications. Samples (20 g) of homogenized freeze-dried liver were mixed with 80 mL K_2_HPO_4_ buffer solution (50 mM, pH 8.0) for 30 s (Scientz Co., Ningbo, China) and then stirred at 4 °C for 1 h. The mixed solution was centrifuged (GL21M, Hunan Kaida Scientific Instrument Co., Ltd., Chenzhou, China) two times for 20 min at 10,000× *g* and 4 °C. Then, the fat and sediments were removed, and the liquid in the middle layer was collected. The collected liquid was heated at 60 °C for 20 min and centrifuged several times at 10,000× *g* for 20 min (4 °C). Then, 50% saturated (NH_4_)_2_SO_4_ was added into the supernatant and kept for 12 h at 4 °C to precipitate the protein. When the sediment appeared, the mixture was centrifuged at 10,000× *g* at 4 °C for 20 min. The supernatant was removed, and the red precipitate was collected. The precipitate was dissolved in buffer solution (50 mM Tris–HCl, pH 7.5) and dialyzed in pure water for 18 h using dialysis membranes of molecular weight within 10 kDa (width is 25 mm, Beijing Baiolibo Technology Co., LTD., Beijing, China). The dialyzed solution was separated and purified. Ferritin was purified by anion exchange chromatography in a DEAE-Sepharose Fast Flow (Shanghai Yuanye Biotechnology Co., Ltd., Shanghai, China) column. The column (φ1.6 cm × 20 cm, Beijing Baiolibo Technology Co., Ltd., Beijing, China) was equilibrated with 50 mM Tris–HCl (pH 9.0), and the ferritin was purified by gradient elution with 50 mM Tris–HCl (pH 9.0) containing NaCl (0–0.2 M). The flow rate was 1.5 mL/min. Finally, the absorbance of eluate was measured at 280 nm. 

### 2.3. Thermal Treatment of Ferritin

The protein concentration was determined by the Lowry method. The protein concentration was adjusted to 0.225 mg/mL, and 70 mL was dispensed into twenty centrifuge tubes. Subsequently, the samples were heated in the bath water at different temperatures (60 °C, 70 °C, 80 °C, 90 °C, and 100 °C) for 10 min, and untreated ferritin was used as a control [22].

### 2.4. Determination of the Solubility and Iron Content of NPLF 

The solubility of ferritin was measured according to a previously reported method by Zhang et al. [23]. Five milliliters of each thermally treated sample were centrifuged at 5000 rpm for 10 min, and the supernatant was retained to determine the protein concentration by the Lowry method. The protein solubility was calculated by the ratio of the protein concentration of the supernatant to the total protein concentration.

The iron content in the supernatant was determined by the Ferrozine method, with minor modifications [24]. The specific steps were as follows: 750 µL of sample solution was accurately measured and combined with 250 µL 10% trichloroacetic acid. The mixture was centrifuged at 4000× *g* for 1 min. Then, 650 µL of the supernatant was collected and combined with 100 µL saturated ammonium acetate, 62.5 μL 0.12 M ascorbic acid, and 62.5 μL 0.25 M Ferrozine. The mixture was diluted to 1 mL with distilled water. After reacting for 4 h, the absorbance was measured at 562 nm by a Cary 50 spectrophotometer (Shanghai Spectrum Instrument 175 Co., Ltd., Shanghai, China). The working curve was plotted with different concentrations of FeSO_4_·7H_2_O standard solution.

### 2.5. Protein Gel Electrophoresis

Protein sodium dodecyl sulfate–polyacrylamide gel electrophoresis (SDS–PAGE) and native polyacrylamide gel electrophoresis (Native-PAGE) were performed using the method described by Kong et al., with slight modifications [25]. SDS–PAGE was performed using 15% polyacrylamide separating gel and 5% polyacrylamide stacking, and a constant current of 30 mA was applied. The samples (10 μg) were added to the gel plate for electrophoresis (Bio-Rad Laboratories, Hercules, CA, USA). The conditions of electrophoresis were as follows: 30 mA for 40 min for running the stacking gel, and 40 mA until the dye front reached the bottom of the gel. Band intensities were analyzed by Image J (version 1.7.0).

Native-PAGE was performed using 5–20% gel and 10 mA (Bio-Rad Laboratories, Hercules, CA, USA). After electrophoresis, gels were stained with Coomassie brilliant blue R-250 for 40 min and decolorized with 25% methanol and 7.5% acetic acid overnight. Then, the purity of the sample was evaluated, and the molecular weight of the target protein was estimated by observing the gel electrophoresis. Native-PAGE markers were a mixture of five proteins, including thyroglobulin 669 kDa, ferritin 440 kDa, recombinant protein 228 kDa, bovine serum albumin 66 kDa, and ovalbumin 45 kDa. 

### 2.6. Dynamic Light Scattering Experiments

The particle size changes of ferritin under different temperature treatments were measured by a dynamic light scattering instrument (LS Instruments AG, Fribourg, Switzerland) at 25 °C according to Yang et al. [26]. The scattering angle was 90 °C, and the measurement time was 1 min. The average value of three repeated measurements was calculated and the graph was drawn using Origin 8.5.

### 2.7. Circular Dichroism Measurements

The secondary structure of ferritin was studied using a MOS-450 circular dichroism spectrometer (Biologic, Rennes, France). This method was done according to a previous report [19]. Measurements were performed at the range of 190–260 nm at 25 °C, and a quartz absorption cell with an optical path length of 1 mm was used. Each treatment was repeated three times, and the average value was calculated. The proportion of each secondary structure (α-helix, β-sheet, β-turn, and random coil) was analyzed by the DichroWeb Server http://dichroweb.cryst.bbk.ac.uk/html/home.shtml (accessed on 21 December 2021) [27].

### 2.8. UV-Visible Absorption Spectrum

UV–Visible absorption spectrum analysis (Cary 50, Shanghai Spectrum Instrument Co., Ltd., Shanghai, China) was carried out according to a previous method, with slight modification [28]. All samples were processed, the blank control was Tris–HCl buffer solution. The ultraviolet spectra were collected in the range of 200–400 nm. The second-order derivation was carried out by using the result of the ultraviolet spectrum.

### 2.9. Intrinsic Fluorescence Measurements 

The intrinsic fluorescence spectroscopy of samples was performed by the method of Zhu et al. [29], with small modifications. The fluorescence data were collected at 300–500 nm, with an excitation wavelength of 290 nm (970CRT, INESA Analytical Instrument Co., Ltd., Shanghai, China), and the slit-width was 5 nm. 

### 2.10. Transmission Electron Microscope Experiments

The sample preparation method of the transmission electron microscope was modified with reference to a previously reported method [30]. Different samples were diluted with 50 mM Tris–HCl (pH 7.0) buffer and then placed on a copper net coated with carbon film for 10 min. After the samples were dried, 2% uranyl acetate was added for 5 min, and the excess solution was removed with filter paper. The samples were observed and imaged by a transmission electron microscope at 80 kV (HT7700, Hitachi, Tokyo, Japan).

### 2.11. Statistical Analysis

All graphs were drawn with Origin 8.5 software, and SPSS 17.0 software was used to perform significant difference analysis. All experiments were repeated three times, and the results are expressed as the mean ± standard deviation (SD). The significant level was set at *p* < 0.05.

## 3. Results and Discussion

### 3.1. Characterization and Thermal Stability of NPLF

Native-PAGE was used to identify the purity of the ferritin samples, and the electrophoretogram is shown in Figure 1A. NPLF mainly consisted of single ferritin (about 450 kDa) and multimeric ferritin. The results are consistent with the findings of Kong et al. who reported *Dasyatis akajei* ferritin isolated from liver, with a molecular weight of 400 kDa [25]. Li et al. found oyster ferritin was also composed of multimer and individual ferritin (480 kDa) [19]. With increasing temperature, the gray values of individual ferritin and polymer bands were gradually reduced. This result indicates that ferritin depolymerized with increasing temperature. At 100 °C, almost all of the bands vanished, suggesting that the protein was totally denatured.

The electropherogram (Figure 1B) showed a single band with a molecular weight of roughly 20.4 kDa, which was a subunit of the NPLF. The results is consistent with previous findings [31]. Chen et al. discovered that *Sphyma zygaena* liver ferritin (SZLF) also had only one subunit, with a molecular weight of 20 kDa [31]. After treatment at 60–80 °C, there were no obvious changes in the bands of the samples. This result indicates that ferritin did not aggregate under this condition and maintained its structural stability. At 90 °C and 100 °C, the ferritin subunit was obviously degraded. After analysis by Image J (Figure 1C), the gray value in the 90 °C and 100 °C groups decreased by 26.97% and 58.17%, respectively, compared with the control. The gray value of electrophoretic bands was considered a reference for the change in protein content. Thus, we speculate that NPLF formed insoluble aggregates, which led to a decrease in protein solubility and the gray value of the electrophoresis band, as the temperature reached 90 and 100 °C.

### 3.2. Solubility and Iron Content of NPLF

Excessive temperature leads to protein denaturation and even degradation. Therefore, different temperature treatments were used to study the relationship between temperature and physicochemical properties of ferritin. As shown in Figure 2, the solubility of NPLF decreased continuously as the temperature increased. The present result is supported by the work of Sorgentini et al., who showed the content of the insoluble protein fraction in soy protein was greater at 100 °C than that at 80 °C [32]. When the temperature was <80 °C, the protein solubility was not significantly different (*p* > 0.05). However, significant differences (*p* < 0.05) were found when the temperature exceeded 80 °C. According to a previous study, the protein was easily decomposed to produce small soluble substances when the heating temperature was low [32]. As the temperature increased, the molecular structure of protein unfolded, and the hydrophobic groups were exposed. These exposed groups will collide continuously, cross-linking with each other to form insoluble aggregates under the combined influence of various forces. This results in a decrease in solubility [28]. Thus, decreased protein solubility of NPLF can be attributed to the generation of insoluble aggregates by thermal induction.

In addition, as shown in Figure 2, no obvious changes were observed in the iron content of NPLF at 60 °C and 70 °C (*p* > 0.05), compared with the control. At 80 °C, the iron content of ferritin was significantly decreased (*p* < 0.05), with a loss of 10.3%, from 188 ± 9.54 μM to 168.67 ± 3.79 μM, compared with control. As the thermal treatment temperature increased to 90 °C and 100 °C, the iron content of NPLF decreased by 29.6% and 75.5%, respectively, compared with the control group. This trend is consistent with the finding of Tang et al., who found pea seed ferritin lost approximately 54% of the total iron during thermal treatment at 90 °C [18]. There are multiple two-fold, three-fold, and four-fold channels in ferritin. These channels are responsible for the material exchange between ferritin and the external environment, such as iron ions, oxygen molecules, and other molecules or ions [7]. The three-fold channels are the main channels for iron entering and leaving the protein. The increase in temperature caused the expansion of the three-fold channels, promoting the leakage of iron from the inside of the protein and decreasing the iron content significantly [33]. In addition, the reason no significant difference in the iron content of NPLF at 60–80 °C was observed is because the expanded channels were reassembled into a complete spherical structure after treatment, which prevented the continuous leakage of iron [34]. Therefore, it can be inferred that the content of iron in ferritin may be related to the structural integrity and self-assembly ability of ferritin. 

### 3.3. Monodispersity of NPLF under Thermal Treatment

The particle size of protein can reflect the aggregation degree of protein. The variation in particle size of NPLF with thermal treatment temperature is shown in Figure 3. The particle size of untreated ferritin (25 °C) had a single peak distribution at 0–10 nm, and the hydraulic radius was 13.41 ± 1.64 nm (Figure 4). When the treatment temperature was 60 °C and 70 °C, the particle peak shifted slightly to the right at 0–10 nm, and the hydraulic radius increased to 18.43 ± 0.82 and 19.43 ± 1.45 nm, respectively. Yet, a single peak distribution was still apparent. The peak had a bimodal distribution at 80 °C, with one peak at 0–10 nm and the other peak at 10–100 nm, indicating that the particle size had a wide range. As the thermal treatment temperature increased to 90 °C and 100 °C, the area of the particle size distribution of NPLF increased at 100 nm, and the hydraulic radius also increased significantly (*p* < 0.05) to 28.03 ± 0.54 nm and 43.99 ± 3.24 nm, respectively. The results are similar to the particle size distribution of red bean seed ferritin studied by Meng et al. [28]. Yang et al. indicated the particle size of ferritin showed a single peak distribution, which proved it had preferable monodispersity [26]. The monodispersity of NPLF was not greatly affected by temperature (<80 °C). This is due to the existence of a large number of salt bridges and hydrogen bonds between ferritin subunits, which impart good thermal stability. However, the particle size increased with increasing temperature (>80 °C), likely due to the aggregation of proteins or the generation of aggregates with low solubility. Another reason for the increase in particle size may be that when the temperature of the protein solution is higher than the denaturation temperature, the proteins expand and then bind to each other, resulting in molecular aggregation [35]. Li et al. reported a similar trend in recombinant oyster ferritin under thermal treatment. In the present study, these results are consistent with the analysis of protein solubility (Figure 2), which further explain the aggregation of ferritin upon heat treatment, resulting in a decrease in solubility [19]. A previous study reported ferritin subunits formed hollow cage-like 24-mers by self-assembly. Only monodisperse protein cages had self-assembly properties, and when aggregation occurred, the ability of ferritin to self-assemble was no longer possible [36]. 

Here, we demonstrate thermal treatment, especially below 80 °C, did not affect the self-assembly ability of ferritin. According to a previous report, thermal treatment at 60 °C expanded the ferritin channels and captured the bioactive molecule rutin [34]. Upon a decrease in temperature to 20 °C, soybean seed ferritin restored its complete spherical structure through self-assembly, which retained rutin within the ferritin cage. This research also supports our results.

### 3.4. Secondary Structure Changes of NPLF under Thermal Treatment

Ferritin as a new type of iron supplement, and its bioactive function depends on its unique cage structure and chemical properties. Therefore, we explored the effect of temperature on the changes in its structure. The CD spectra of NPLF under different thermal treatments are shown in Figure 5A. Ferritin showed an obvious positive peak around 190 nm, which might be caused by the coincidence of the characteristic peak of α-helix at 192 nm and the characteristic peak of β-sheet at 185–200 nm. This indicates the protein structure contained α-helices and β-sheets, similar to plant, mammal, and microbial ferritin [33]. In addition, there were two obvious negative peaks between 200 and 210 nm, which indicate ferritin was mainly composed of α-helix structures [20]. DichroWeb was employed to analyze the relative percent content of the NPLF secondary structure at different temperatures, and the results are shown in Table 1.

In untreated NPLF, the contents of α-helix, β-sheet, β-turn, and random coils were 62.83%, 22.17%, 7.67%, and 7.33%, respectively. In this study, as thermal treatment temperature increased, the contents of α-helix decreased significantly (*p* < 0.05), but the contents of β-turn and random coils increased gradually. In protein molecules, α-helices and β-sheets can form a compact structure, while the conformational stability and compactness of random coils are poorer. In addition, the contents of α-helix are related to the stability of the hydrogen bonds between polypeptide chains [19]. The decrease in the α-helix content of ferritin under high-temperature treatment was attributed to the loosening of the hydrogen bonds that maintain the α-helix structure. It can be presumed that the decrease in the α-helix content led to the change in the rigid and elastic structure of NPLF, which affected the stability of the entire protein. When the temperature was above 80 °C, the proportion of β-sheet content increased. The β-sheets are usually related to the aggregation state of proteins. The larger the proportion of β-sheets, the higher the aggregation degree of the protein [2]. In general, the α-helix structure depends on intramolecular hydrogen bonds of peptide chains, while the β-sheet structure depends on intermolecular hydrogen bonds of peptide chains [37]. Therefore, the interaction between multiple subunits of ferritin led to the formation of intermolecular β-sheet structures [38] and the generation of aggregates under this condition. In other words, treatment at 90 °C and 100 °C induced the recombination of intramolecular/intermolecular hydrogen bonds of ferritin, which changed the protein structure from ordered to disordered. At 90 °C and 100 °C, the content of random coils increased significantly (*p* < 0.05). A similar trend in β-lactoglobulin was reported by Loveday [11]. This phenomenon can be attributed to the expansion and disintegration of NPLF cages under high temperature treatment, leading to a change in the ferritin secondary structure [39]. 

### 3.5. Tertiary Structure Changes in NPLF under Thermal Treatment

#### 3.5.1. UV–Visible Absorption Spectrum Analysis

Generally, variations in UV spectra are caused by the number of chromogenic amino acids, which expose the protein surface. Tryptophan (Trp), tyrosine (Tyr), and phenylalanine (Phe) have different UV absorption spectra because of their different chromophores. Trp and Tyr have an absorption peak around 280 nm, and Phe has an absorption peak around 257 nm [40]. As shown in Figure 5B, the UV spectra of samples had peaks around 280 and 257 nm, indicating Trp, Tyr, and Phe were present in ferritin after different temperature treatments. As the temperature increased, the absorption value of ferritin at 280 nm gradually decreased and red-shifted. This was attributed to the Trp environment becoming more hydrophobic as the temperature increased. The Trp residues in ferritin are mainly located in four-fold channels, and these structures are sensitive to environmental conditions such as temperature changes [41]. Thus, we speculate that the variation in the UV spectra was caused by the change in the Trp microenvironment along the four-fold channels of ferritin.

UV secondary derivatization spectra are helpful in observing the state and contents of aromatic amino acids in proteins. To accurately analyze and compare the microenvironment changes of Trp, Tyr, and Phe in different samples, the second-order derivation of the UV scanning spectrum was performed [42]. In Figure 5C, the UV secondary derivative diagram of untreated NPLF had a series of positive peaks, such as 254, 274, 283, and 295 nm. The negative peaks were at 258, 271, 278, and 289 nm. After thermal treatment, the optical density of ferritin in the range of 250–300 nm changed, and the positions of the peaks and valleys shifted. At 60 °C and 70 °C, a blue shift in λ_max_ of Phe appeared (the wavelength of the maximum absorption peak of the absorption band moved towards the short-wave direction). After the sample was heated (>80 °C), the peaks and valleys corresponding to Phe at 254, 258, and 271 nm red shifted (the wavelength of the maximum absorption peak of the absorption band moved towards the long-wave direction) and exhibited a decrease in intensity. Similarly, the peaks of Tyr and Trp at 283 and 290 nm, respectively, and the valley at 289 nm all showed a red shift in λ_max_. The results indicate that when the temperature was higher than 80 °C, the microenvironment of Trp, Phe, and Tyr became hydrophobic. In the second derivative UV spectra, the position of the aromatic amino acid peak and the ratio of peak-to-valley longitudinal distance reflected its exposure [42]. In this study, the changes in the microenvironment of Tyr were expressed by ‘r’ (r = a/b). When the temperature was lower than 80 °C, the value of r increased and then decreased with increasing temperature. The above results indicate the tertiary structure of ferritin unfolded at 80 °C, and the hydrophobic amino acids buried in the protein moved to the protein shell, which led to the exposure of these amino acids to the hydrophilic environment [43]. However, with the increase of thermal temperature (>80 °C), the structure of ferritin was further destroyed, and excessive hydrophobic groups were exposed, resulting in the aggregation and precipitation of protein [15]. These results are consistent with the results of NPLF solubility and electrophoresis. 

#### 3.5.2. Intrinsic Fluorescence Analysis

Because of its high sensitivity, the fluorescence spectrum is an important means widely used to study the changes in the protein tertiary structure. The emission fluorescence spectrum results of NPLF under different thermal treatment conditions are shown in Figure 5D. Trp, Tyr, and Phe residues can emit fluorescence under UV light at 270–300 nm. Trp has the highest fluorescence intensity; thus, it can be used to identify the conformational changes of protein [15]. At an excitation wavelength at 290 nm, it reflects the fluorescence spectrum with Trp as the emission group. In the wavelength range of 300–500 nm, the fluorescence peak of untreated NPLF was 351.7 nm. As the temperature increased, the fluorescence emission peak of Trp was quenched. This might be the result of unfolding of the ferritin structure, which transferred the hydrophobic amino acids from the inside of the protein to the outside of the protein [5]. In addition, the fluorescence intensity of ferritin increased at 60 °C and 70 °C, and the maximum emission wavelength shifted to 352.4 and 352 nm, respectively. The maximum absorption peak of the endogenous fluorescence spectrum shifted a longer wavelength (red-shift) when tryptophan migrated to the hydrophilic environment. It is possible that tryptophan was exposed to the hydrophilic environment during the reassembly of ferritin, indicating ferritin self-assembled at this temperature and stabilized its structure [44]. Thus, we speculate that mild heat treatment (<80 °C) could regulate the self-assembly activity of ferritin by transferring hydrophobic amino groups. Compared with the untreated samples, the maximum absorption peak λ_max_ of ferritin exhibited a blue shift to 348 nm at 90 °C. It is possible that the amino acid side chains of ferritin were buried in the solvent, which made tryptophan move to a hydrophobic environment. The increase in hydrophobic interactions disrupted intermolecular and intramolecular hydrogen bonds and made the structure more disordered [45]. However, proteins aggregate and form insoluble clusters at high temperatures. This aggregation prevented the action of tryptophan residues in polar environments. Li et al. found that the fluorescence intensity of oyster ferritin decreased with increasing heating temperature, and the maximum absorption peak of the protein sample had an obvious blue shift, which was in agreement with our study [19]. The results show that the secondary and tertiary structures of protein change during heat treatment, promoting the formation of protein aggregates [15]. 

### 3.6. Microstructure Changes in NPLF under Thermal Treatment

Figure 6 shows the morphological changes of NPLF at different temperatures. Ferritin has a hollow cage-like structure, and after negative staining with uranyl acetate, uranium flowed into the cavity through the ferritin channels, showing many uranium-containing nuclei. It can be seen from Figure 6 that NPLF had the same protein shell structure as other known ferritins and a uniform distribution [46]. Compared with the untreated samples, ferritin still retained its complete protein cage structure and distributed uniformly at 60 °C and 70 °C. At 80 °C, although the ferritin formed small aggregations, it still maintained a complete spherical structure. Thus, we speculate the spherical structure might be due to the self-assembly at this temperature. This led to the partially unfolded structure reassembly, which promoted the aggregation of ferritin and produced the small aggregates [45]. This is consistent with the results of DLS. Thus, mild thermal treatment (<80 °C) not only inhibited the aggregation of ferritin, but also maintained its structural integrity. When the sample was heated to 90 °C, most of the ferritin aggregated. Thermally induced aggregation of globular proteins is mainly caused by the unfolding of the protein structure, and hydrophobic interaction is considered to be the primary mechanism of thermal aggregation [40]. High thermal treatment caused a change in the distribution of ferritin from uniform to aggregate, which was attributed to the exposure of hydrophobic groups under a high temperature treatment [47]. When the temperature reached 100 °C, the protein aggregated completely, and the complete spherical structure disappeared. Ferritin was completely denatured and could not self-assemble at 100 °C. This thermally induced aggregation was irreversible. Tang et al. showed thermal treatment (60–80 °C) prolonged the storage time of pea seed ferritin, inhibited the aggregation, and promoted the dispersion of protein [18]. However, exposure to higher temperatures (90 °C or higher) led to the structural denaturation of ferritin. Hence, the results of TEM also demonstrates that the increase in temperature led to changes in the NPLF structure, which affected its self-assembly characteristics.

## 4. Conclusions

NPLF from liver represents a new source of ferritin. Compared with pea seed ferritin, soy ferritin, and other plant ferritin, it is more stable due to the lack of the extension peptide. In this study, we found that the thermal treatment temperature affected the structure and self-assembly capabilities of ferritin. The usual cage-like structure of liver ferritin was preserved when the treatment temperature was below 80 °C. Under this condition, the expanded channel of NPLF formed a complete spherical structure by reassembly, preventing the continuous leakage of iron. However, as the temperature increased, the expansion of the channel was not recovered and led to a significant decrease in iron content. This indicates that high-temperature treatment (90 and 100 °C) damaged ferritin’s structure. The results of CD spectroscopy and intrinsic fluorescence spectroscopy also supported this finding. Notably, the fluorescence intensity of ferritin increased at 60 °C and 70 °C. We hypothesized that thermal treatment could regulate the self-assembly activity of ferritin by controlling the transfer of hydrophobic amino acids. This novel observation indicates treatment below 80 °C did not affect the cage-like structure and self-assembly ability of ferritin, and aggregation of NPLF was prevented. Thus, NPLF processing temperatures should be kept below 80 °C to further develop the processing and use of ferritin. 

## Figures and Tables

**Figure 1 foods-11-02987-f001:**
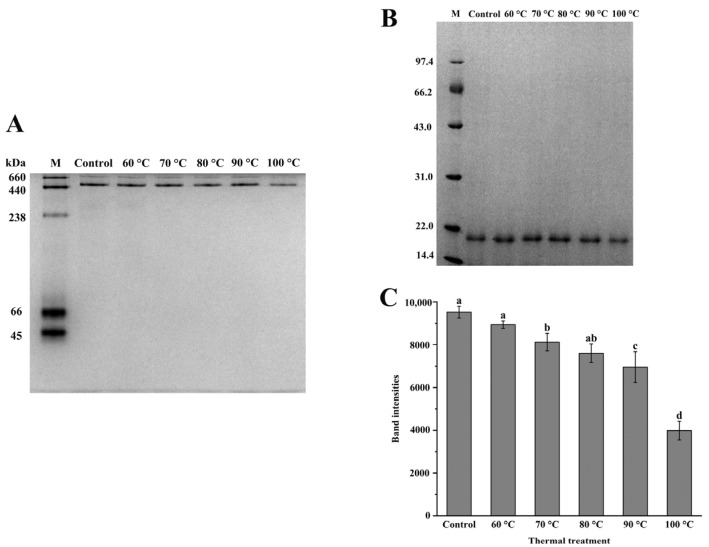
Electrophoresis analyses of northern pike liver (NPLF) at different temperatures. (**A**) Native-PAGE. (**B**) SDS–PAGE. (**C**) Band intensities of NPLF under thermal treatment. Values with different letters are significantly different (*p* < 0.05).

**Figure 2 foods-11-02987-f002:**
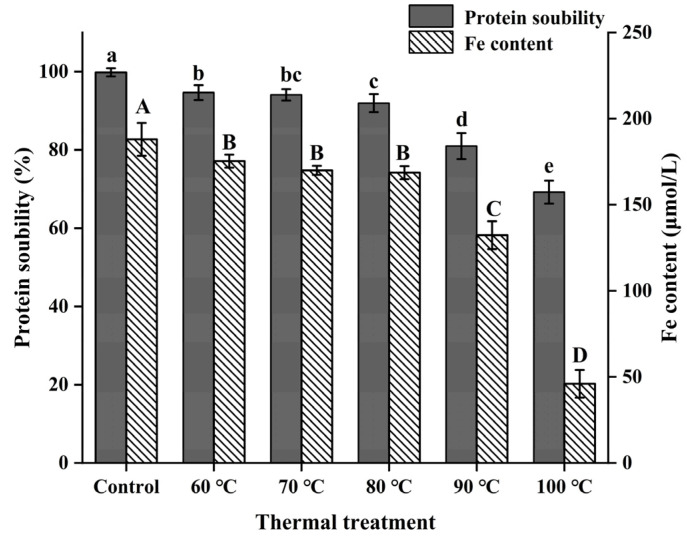
Effects of thermal treatment at different temperatures (untreated, 60, 70, 80, 90, and 100 °C) for 10 min on the solubility and iron content. The differences between the protein concentration group and the iron content group are indicated by lowercase and uppercase letters, respectively (*p* < 0.05). The error bars represent the standard error of the mean.

**Figure 3 foods-11-02987-f003:**
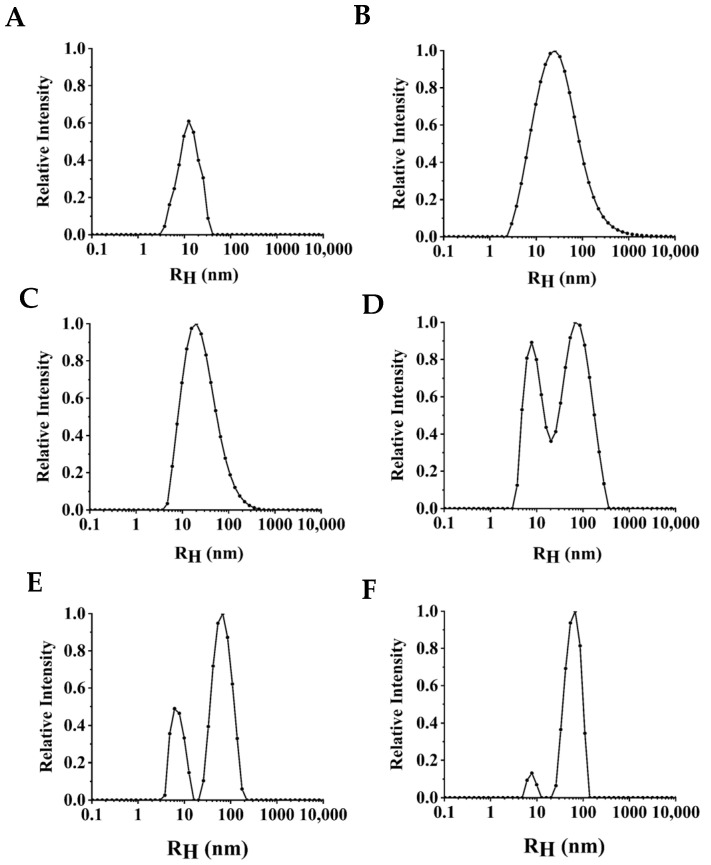
Effects of thermal treatment at different temperatures (untreated, 60, 70, 80, 90, and 100 °C) for 10 min on the particle size distribution of ferritin. Figures (**A**–**F**) are, respectively, control (untreated ferritin) and thermal treatment of 60, 70, 80, 90, and 100 °C.

**Figure 4 foods-11-02987-f004:**
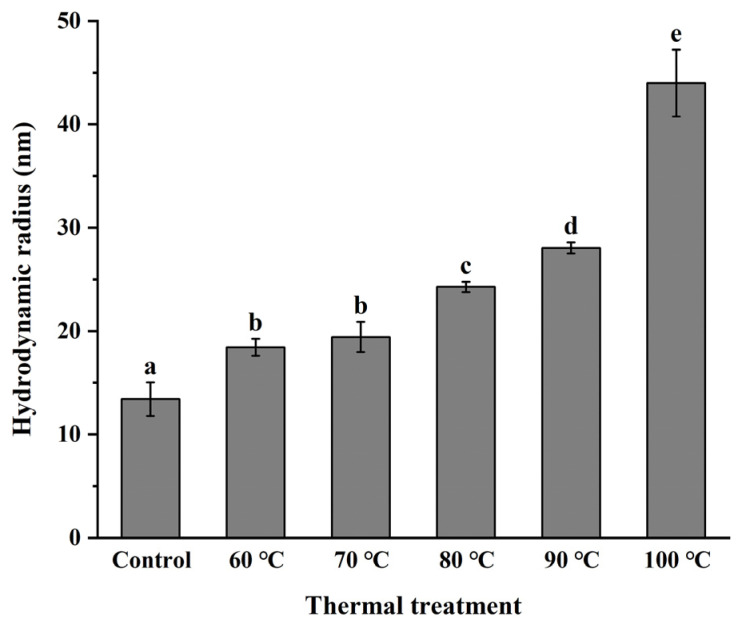
Effects of thermal treatment at different temperatures (untreated, 60, 70, 80, 90, and 100 °C) for 10 min on the hydraulic radius of ferritin. Values with different letters are significantly different (*p* < 0.05).

**Figure 5 foods-11-02987-f005:**
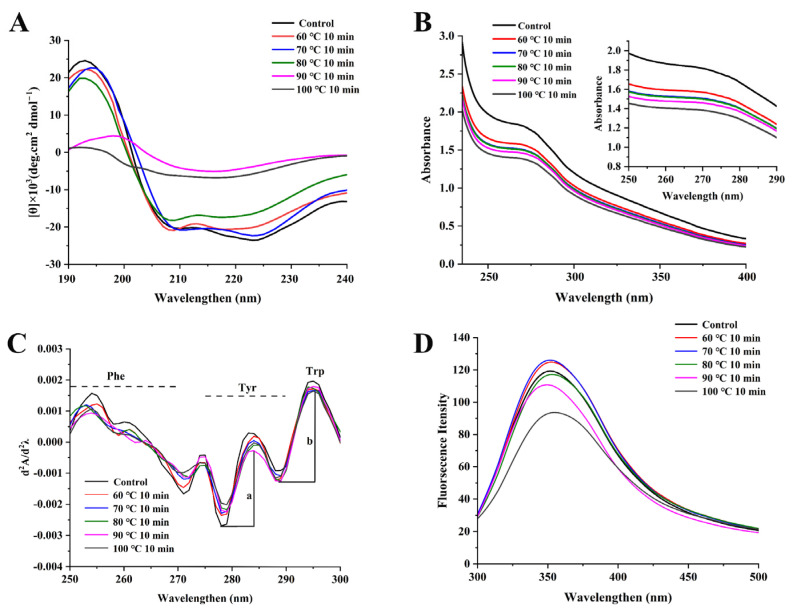
Effects of thermal treatment at different temperatures (untreated, 60, 70, 80, 90, and 100 °C) for 10 min on the structure of ferritin. (**A**) CD spectrum, (**B**) UV absorption spectra, (**C**) second-derivative UV spectra, the letters ‘a’ and ‘b’ represent the peak-to-trough values for the two main peaks, (**D**) intrinsic fluorescence.

**Figure 6 foods-11-02987-f006:**
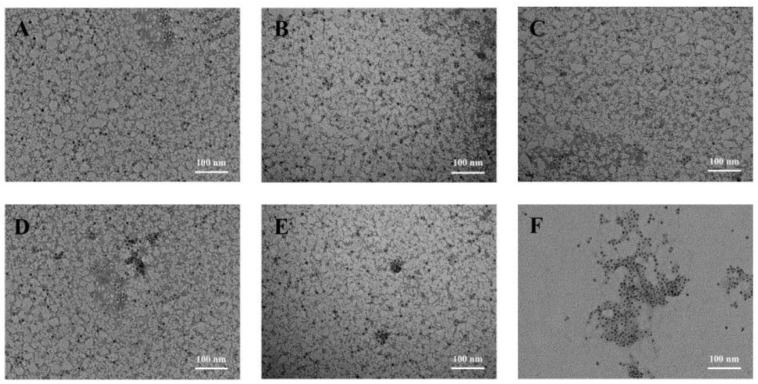
TEM images of NPLF at different thermal treatment temperatures. Figures (**A**–**F**) are control (untreated ferritin) and thermal treatment on 60, 70, 80, 90, and 100 °C, respectively.

**Table 1 foods-11-02987-t001:** Percentage of secondary structure of the northern pike liver ferritin at different temperatures (untreated, 60, 70, 80, 90, and 100 °C).

Treatment Temperature	α-Helix (%)	β-Sheet (%)	β-Turn (%)	Random Coils (%)
control	62.83 ± 0.65 a	22.17 ± 0.50 c	7.67 ± 0.72 f	7.33 ± 1.25 f
60 °C	59.70 ± 0.50 b	17.37 ± 0.95 d	11.33 ± 0.60 e	11.60 ± 1.08 e
70 °C	50.60 ± 1.06 c	22.57 ± 0.85 c	12.93 ± 0.68 d	13.90 ± 0.60 d
80 °C	48.57 ± 0.86 d	15.25 ± 0.46 e	15.34 ± 0.93 c	20.83 ± 0.76 c
90 °C	8.10 ± 0.79 e	36.13 ± 0.84 a	23.43 ± 1.35 b	32.33 ± 0.12 b
100 °C	1.90 ± 0.56 f	27.40 ± 0.20 b	35.66 ± 0.68 a	35.03 ± 0.42 a

All data are the mean ± standard deviation (SD) of three experiments. Different letters above the same column indicate significant difference (*p* < 0.05).

## Data Availability

The data presented in this study are available upon request from the corresponding author.

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
