# Peer review of "Modifications of Thermal-Induced Northern Pike (Esox lucius) Liver Ferritin on Structural and Self-Assembly Properties"

_foods, 2022, doi:10.3390/foods11192987_

Round 1

Reviewer 1 Report

Thank you for the opportunity to provide a peer-review for this interesting manuscript, which reports the results of the investigation of the structural changes in Northern pike (Esox lucius) hepatic ferritin under several thermal conditions (i.e., without thermal treatment, 60, 70, 80, 90 and 100 °C) and the effect of temperature on its self-assembly capabilities, with the ultimate aim of exploring its potential as a food supplement in the fight against anaemia caused by iron deficiency.

As a whole, the manuscript deals with pertinent issues and brings novel data that are properly presented and discussed. Furthermore, the methodological robustness is unequivocal. However, the following suggestions should be taken into account in a subsequent re-submission.

1.      Title: Thermal-induced structural modify and self-assembly characteristics of Northern pike (Esox Lucius) liver ferritin”.Esox Lucius” should be corrected to “Esox lucius”.

2.      “Electrophoresis, dynamic light scattering (DLS), circular dichroism (CD), UV-Visible absorption spectroscopy, fluorescence spectroscopy, and transmission electron microscopy were used to examine changes in ferritin from the liver at 60-100°C (TEM)”. It should be “transmission electron microscopy (TEM) were used to examine (…)”.

3.      At 60°C and 70°C, ferritin's α-helix content was greater. The content of α-helix reduced to 8.10% and 1.90% when the temperature hit 90°C and 100°C, respectively, indicating that the protein structure became loose and lost self-assembly ability.” There should be a space after a number and before °C and units such as mL and min, but not before %. Please apply this correction throughout the document. See the instructions for authors on the journal submission guidelines.

4.      To compensate for the aforementioned shortcomings, ferritin with high iron absorption and low side effects has been proposed as a new iron supplements”. An appropriate reference should be added to support this information.

5.      Ferritin is a kind of storage ferritin consisting of a spherical protein shell and iron core, which widely exists in humans, mammals, plants, fungi, bacteria, and other organisms.” Please rewrite the sentence so that the information is correct and clear.

6.      “Ferritins in animal are mainly found in metabolically active tissues such as the liver and spleen, while ferritins are mainly found in the non-green plastids such as proplastids, amyloids and the top of root”. Please rewrite the sentence so that the information is correct and clear.

7.      “Thousands of iron atoms exist as phosphate minerals iron hydroxide in the inner cavity of ferritin [6]”. Please clarify this sentence.

8.      Besides that, ferritin could withstand higher temperature and extreme pH conditions”. This sentence may be confusing considering the purpose of the work reported in the manuscript. Could the authors please clarify it? Also, an appropriate reference should be added.

9.      “Hence, ferritin could be explored as a safe and effective functional iron supplement”. It would be helpful to add more data to support the safety and effectiveness of ferritin as a food supplement.

10.   “Esox Lucius (weight 900 ± 50 g, length 50 ± 5 cm) were purchased from a local market of aquatic products (Shihezi, China) and were sent to the laboratory using plastic bags containing of suitable oxygen within 30 min”. “Esox Lucius” should be corrected to “Esox lucius”.

11.   “Acrylamide, 2-Hydroxy-1-ethanethiol, sodium lauryl sulfate (SDS), tetramethyl ethylenediamine (TEMED), ammonium persulfate (APS), ethylenediamine tetra acetic acid (EDTA), Tris(hydroxymethyl)methyl aminomethane were all purchased from Beijing Biotopped, Ltd. (Beijing, China)”. “2-Hydroxy-1-ethanethiol” should be corrected to “2-hydroxy-1-ethanethiol” and “Tris(hydroxymethyl)methyl aminomethane” should be corrected to “tris(hydroxymethyl)methyl aminomethane”.

12.   “The mixed homogenate was centrifuged two times at 10,000 g for 20 min at 4°C using a homogenizer (…), filtrated to remove the fat, and the precipitate collected the liquid in the middle layer”. Could the authors please double-check this sentence? It is not clear enough.

13.   Then dissolved in buffer solution (50 mM Tris-HCl, pH 7.5) and dialysis (10 kDa) with pure water for 18 h.” Could you please provide more details about the dialysis process, namely the membrane type used?

14.   “Ferritin was purified by anion exchange chromatography in a DEAE-Sepharose Fast Flow column”. Could you please provide more details about the column (e.g. supplier company; column height and diameter)?

15.   “After reaction 4 hours, measured the absorbance at 562 nm”; “The particle size changes of ferritin under different temperature treatments were measured by a dynamic light scattering instrument at 25°C according to Yang et al. [21] reported. The name of the supplier company, city and country should be mentioned for chemicals and equipment. This is not always complete on some items and sometimes it is not provided at all. 

16.   “The parameter of scattering angle is 90°C, and the average value was taken for analysis after three repeated measurements, and the graph was drawn with origin 8.5.” vs. “The changes of secondary structures (α-helix, β-sheet, β-turn and random coil) proportion of the samples were analyzed by the DichroWeb Server [22], which was located at Internet (…)”. The English language variant chosen (UK English or US English) should be used consistently, I recommend that this aspect be checked and corrected. Overall, the document could benefit if revised by a native English speaker.

17.   “This method was done according to a previously reported14.” The reference needs to be corrected. See the instructions for authors on the journal submission guidelines.

18.   3. Results”. It should be “Results and Discussion”.

19.   “The electropherogram shows a single band with a molecular weight of roughly 20.4 kDa, which could be a subunit of the northern pike liver ferritin, as shown in Figure 1B.” Abbreviations should be properly defined the first time they are mentioned in the text and also the first time they are mentioned in a table or figure. Henceforth, they should be used instead of the complete form. This correction should be applied throughout the manuscript.

20.   “Figure 1”. From my point of view, the temperature at which the different treatments took place should be indicated more clearly in the subfigures, similarly to what the authors did in Figure 2, instead of indicating 1,2,3, etc…

21.   “It also could be seen from Figure 2 that compared with the control sample, there was no obvious difference in the iron content of NPLF after thermal treatment at 60°C and 70°C. At 80°C for 10min resulted a significant decrease (p < 0.05) of iron contents, with a loss of 10.3% from (188 ± 9.54) μM to (168.67 ± 3.79) μM”. The observations reported in the text do not seem entirely consistent with the data presented in the graph. The bars at 70°C and 80 °C appear to be identical. Could the authors please double-check this?

22.   “It could be seen that the particle size of untreated ferritin (25°C) had a single peak distribution at 0-10 nm and the hydraulic radius was (13.41 ± 1.64) nm (Fig. 4)”. It should be “13.41 ± 1.64 nm”. Please apply this correction throughout the document.

23.   Figure 5A. Please add the appropriate units to the x-axis.

24.   Table 1.All data are the mean ± standard deviation (CD) of three experiments”. “CD” should be corrected to “SD”.

25.   References need to be carefully verified (e.g., i) “1. Yokoi, K.; Konomi, A. Iron deficiency without anaemia is a potential cause of fatigue: meta-analyses of randomised controlled trials (…)”, the first letter after the comma should be capitalised; ii) “3. Tolkien, Z.; Stecher, L.; Mander, A. P.; Pereira, D. I. A.; Powell, J. J. Ferrous Sulfate Supplementation Causes Significant Gastrointestinal Side-Effects in Adults: A Systematic Review and Meta-Analysis (…)”, except for the first letter of the first word of the title, the others should not be capitalised; iii) “10. Chang, P.; Xie, Y. Y.; Wang, H.; Xia, X. F. Effects of Heat Treatment Temperature and Time on Thermal Aggregation Behavior of Myofibrillar Proteins from Mirror Carp (Cyprinus carpio) (…)”, species names should be in italics. Please see the instructions for authors on the journal submission guidelines.

Reviewer 2 Report

The manuscript by Zhang et. al was designed to study the structural modifications and the relevant self-assembly characteristics of liver extracted from Northern pike. My general impression of the manuscript is not very enthusiastic. The manuscript’s major aim was to study the structural modifications and the relevant self-assembly characteristics of the liver extracts of the pike fish. Still, it did not relate to why studying these experiments would further help the science/food research community to establish the animal-derived (such as this study) as an iron supplement. The research was based on the various experimental setups that were already established and published. In my opinion, it was a routine research paper that was also partially relevant to the journal “foods”. I am not sure if someone could consider the liver extracts containing iron supplements as “food” but rather “nutrients “; hence, MDPI’s journal nutrients could perhaps serve as that platform.

Please also note that it was tough to review this paper because of the hard-to-express English grammar and this was added by the fact that the manuscript DID NOT have sentence numbering. Without numbering it is extremely hard to pinpoint every single sentence.

I am going to write some of the concerns here;

1. Abstract mentions, “it is a new type of natural iron supplement”. Please note that the research on Ferritin has been going on for decades. If there are some new elements that suggest those should have been clearly mentioned.

2. Abstract, “….microscopy were used to examine changes in ferritin from the liver at 60-100°C (TEM).” Why TEM is in the brackets here? It should be next to the instrumental technique.

3. There were bad instances of grammar throughout the manuscript. A wrong blend of past, and present tenses and somewhat “different” use of vocabulary and words. I am only going to relate a few for example, the title of the manuscript, the introduction, 2nd paragraph the sentence stating, “ Ferritin is a kind of storage ferritin consisting of a spherical protein shell and iron..”?

Same paragraph, “Ferritins in animals are mainly found in metabolically active tissues such as the liver..” Please choose either its singular or plural.

Page 2 first paragraph, “When iron concentrations in the environment are low, with the help of reductants ferritin will reduce Fe (III) to Fe (II)” which reductants?

The same paragraph, “Besides that,”….. needs grammar correction

Next sentence, “previous studies have shown that ferritin has low sensitivity to…”

“Research on the effect of thermal treatment on protein structure and physicochemical properties have attracted more and more attention”

Page 2, second paragraph “At the temperature of 135-140°C, the monomer conformation of β-lactoglobulin rearranges,” Either you pick past or present or present indefinite tenses.

Same page “Studies had shown that animal ferritin is more stable than plant ferritin due to lack…” etc.

Page 5, the last paragraph needs correction as well.

As mentioned above these are only a few bad instances of grammar. There were errors throughout the manuscript. At least every other sentence needs revising the English. I highly recommend a native English-speaking colleague would help the authors with this issue.

4. Another major concern here is the absence of declaration from the authors on the subject of “Research and Publication Ethics”.

I am going to copy and paste here in bold font, “Authors should particularly ensure that their research complies with the commonly-accepted '3Rs [1]':

  • Replacement of animals by alternatives wherever possible,
  • Reduction in number of animals used, and
  • Refinement of experimental conditions and procedures to minimize the harm to animals.

Authors must include details on housing, husbandry and pain management in their manuscript.

For further guidance authors should refer to the Code of Practice for the Housing and Care of Animals Used in Scientific Procedures [2], American Association for Laboratory Animal Science [3] or European Animal Research Association [4].

If national legislation requires it, studies involving vertebrates or higher invertebrates must only be carried out after obtaining approval from the appropriate ethics committee. As a minimum, the project identification code, date of approval and name of the ethics committee or institutional review board should be stated in Section ‘Institutional Review Board Statement’. Research procedures must be carried out in accordance with national and institutional regulations. Statements on animal welfare should confirm that the study complied with all relevant legislation. Clinical studies involving animals and interventions outside of routine care require ethics committee oversight as per the American Veterinary Medical Association. If the study involved client-owned animals, informed client consent must be obtained and certified in the manuscript report of the research. Owners must be fully informed if there are any risks associated with the procedures and that the research will be published. If available, a high standard of veterinary care must be provided. Authors are responsible for correctness of the statements provided in the manuscript.

If ethical approval is not required by national laws, authors must provide an exemption from the ethics committee, if one is available. Where a study has been granted exemption, the name of the ethics committee that provided this should be stated in Section ‘Institutional Review Board Statement’ with a full explanation on why the ethical approval was not required.

If no animal ethics committee is available to review applications, authors should be aware that the ethics of their research will be evaluated by reviewers and editors. Authors should provide a statement justifying the work from an ethical perspective, using the same utilitarian framework that is used by ethics committees. Authors may be asked to provide this even if they have received ethical approval”.

In my opinion, the results and findings in this research paper were NOT extraordinary or entirely new e.g., refinement of experimental conditions” or where the benefits of the reported research were far beneficial. The work rather re-iterated the previous findings of Meng et al., Yang et al., Kong et al., Chen et al., Tang et al., and findings were not exceptionally well worth the cause of killing living animals. But this last sentence is my opinion. 

5. I would highly recommend deleting the use of unprofessional sentences such as, “The fish were killed after hitting the head with a wooden stick and took out the liver, homogenized with crusher,…”

Regarding this, the authors did not state why the “living” samples were necessary. I understand if organisms were picked from frozen food sections, it would have affected their proteins but why not establish a connection with the local fisheries dept.? Also, why this particular fish was chosen?

6. The authors claimed that less research has been done on marine/aquatic organism. This is only partly true (e.g., J. L. Miguel, M. I. Pablos, M. T. Agapito, and J. M. Recio. Isolation and characterization of ferritin from the liver of the rainbow trout (Salmo gairdneri R.). Biochemistry and Cell Biology. 69(10-11): 735-741. https://doi.org/10.1139/o91-111; Rosaria Scudiero, Maria Grazia Esposito, Francesca Trinchella, Middle ferritin genes from the icefish Chionodraco rastrospinosus: Comparative analysis and evolution of fish ferritins, Comptes Rendus Biologies,Volume 336, Issue 3, 2013,Pages 134-141; Geetha C, Deshpande V. Purification and characterization of fish liver ferritins. Comp Biochem Physiol B Biochem Mol Biol. 1999 Jul;123(3):285-94. doi: 10.1016/s0305-0491(99)00072-3. PMID: 10481257 etc.,)

6. Correct the scientific name of the organism its wrong throughout the manuscript.

7. Please provide more information and details on the self-assembly ability of ferritin. How was this studied? Please provide details.

8. Figures 3 and 5 are very hard to read. Please provide higher resolution figures.

9. Section 3.5, what is the meaning of “appeared red-shifted”? Please revise this and similar sentences (there are more instances).

10. Fig.6. The TEM images were of poor resolution and quality and were not helpful/ Also, the fig. caption should say the TEM images of the sample.

11. Please elaborate more on the current results rather than the previously done (similar) work. Focus more on what makes this work stand out and NOT done previously.

12. The conclusions ended abruptly and did not justify the case and related how these results could help further in the future for using Ferritin as an Iron supplement. How these studies can help in choosing Ferritin as an Iron supplement.

Round 2

Reviewer 2 Report

I have reviewed the revised version of the manuscript by Zhang et.al and regret to state that the manuscript has not been revised sufficiently. Although some of the concerns have been resolved even in doing so, more errors have been made/added.

The recently added sentences seemed to have so many errors. This paper needs to be REWRITTEN. I am going to add a few bad instances of grammar; Lines 11-14, 42, 46-48, Lines 50-60 (the whole paragraph that has been recently added), Lines 105-108, 131-132, 136-137, 168, 189, 203-205 (tenses are not consistent), 212, the last paragraph on page 5, etc. Revision is needed wherever new sentences have been added. Clearly, THIS MANUSCRIPT NEEDS TO BE REVISED BY A NATIVE ENGLISH SPEAKING COLLEAGUE.

Furthermore, as per my concerns regarding the ethics of the research and the absence of the declaration statement, the authors pointed out, “ The modification has been made in (Line 94-95). According to the method of Li et al. (2022) and Deng et al. (2021) reported, it is an acceptable way to treat fish”?

I do not think, I am concerned with the methods of Deng or Li et al. The authors of the current manuscript should provide a statement of a declaration stating that animals were not unnecessarily killed.  

I am not sure if the authors did not get my suggestions clearly. They need to provide an official statement of declaration ATLEAST stating some of the ethics of the research conducted on the vertebrates such as Northern Pike. They need to provide a such statement before the reference section.

ONCE AGAIN, I am going to copy and paste here in bold font, “Authors should particularly ensure that their research complies with the commonly-accepted '3Rs [1]':

    Replacement of animals by alternatives wherever possible,

    Reduction in number of animals used, and

    Refinement of experimental conditions and procedures to minimize the harm to animals.

Authors must include details on housing, husbandry and pain management in their manuscript.

For further guidance authors should refer to the Code of Practice for the Housing and Care of Animals Used in Scientific Procedures [2], American Association for Laboratory Animal Science [3] or European Animal Research Association [4].

If national legislation requires it, studies involving vertebrates or higher invertebrates must only be carried out after obtaining approval from the appropriate ethics committee. As a minimum, the project identification code, date of approval and name of the ethics committee or institutional review board should be stated in Section ‘Institutional Review Board Statement’. Research procedures must be carried out in accordance with national and institutional regulations. Statements on animal welfare should confirm that the study complied with all relevant legislation. Clinical studies involving animals and interventions outside of routine care require ethics committee oversight as per the American Veterinary Medical Association. If the study involved client-owned animals, informed client consent must be obtained and certified in the manuscript report of the research. Owners must be fully informed if there are any risks associated with the procedures and that the research will be published. If available, a high standard of veterinary care must be provided. Authors are responsible for correctness of the statements provided in the manuscript.

If ethical approval is not required by national laws, authors must provide an exemption from the ethics committee, if one is available. Where a study has been granted exemption, the name of the ethics committee that provided this should be stated in Section ‘Institutional Review Board Statement’ with a full explanation on why the ethical approval was not required. If no animal ethics committee is available to review applications, authors should be aware that the ethics of their research will be evaluated by reviewers and editors. Authors should provide a statement justifying the work from an ethical perspective, using the same utilitarian framework that is used by ethics committees. Authors may be asked to provide this even if they have received ethical approval”.

Please modify some of the relevant sentence from here and provide as a declaration statement.

Also, the statement “KILLING THE ANIMALS” is still considered harsh.

Figure 3 has been revised but the quality of figure 5 is still somewhat poor. I am not sure authors cannot provide pictures with better resolution.
